# A New Methodology for Automatic Cluster-Based Kriging Using K-Nearest Neighbor and Genetic Algorithms

**Carlos Yasojima** [1,*,†,‡] , **João Protázio** [2,‡] , **Bianchi Meiguins** [1,‡], **Nelson Neto** [1,‡] and **Jefferson Morais** [1,‡]

1   Faculty of Computer Science, Federal University of Pará, Belém, PA 66075-110, Brazil; bianchi@ufpa.br (B.M.); nelsonneto@ufpa.br (N.N.); jmorais@ufpa.br (J.M.)
2   Faculty of Statistics, Federal University of Pará, Belém, PA 66075-110, Brazil; mprotazio@gmail.com
*   Correspondence: takeshiyasojima@gmail.com
†   Current address: Faculty of Computer Science, Federal University of Pará, Belém, PA 66075-110, Brazil.
‡   These authors contributed equally to this work.

**Abstract:** Kriging is a geostatistical interpolation technique that performs the prediction of observations in unknown locations through previously collected data. The modelling of the variogram is an essential step of the kriging process because it drives the accuracy of the interpolation model. The conventional method of variogram modelling consists of using specialized knowledge and in-depth study to determine which parameters are suitable for the theoretical variogram. However, this situation is not always possible, and, in this case, it becomes interesting to use an automatic process. Thus, this work aims to propose a new methodology to automate the estimation of theoretical variogram parameters of the kriging process. The proposed methodology is based on preprocessing techniques, data clustering, genetic algorithms, and the K-Nearest Neighbor classifier (KNN). The performance of the methodology was evaluated using two databases, and it was compared to other optimization techniques widely used in the literature. The impacts of the clustering step on the stationary hypothesis were also investigated with and without trend removal techniques. The results showed that, in this automated proposal, the clustering process increases the accuracy of the kriging prediction. However, it generates groups that might not be stationary. Genetic algorithms are easily configurable with the proposed heuristic when setting the variable ranges in comparison to other optimization techniques, and the KNN method is satisfactory in solving some problems caused by the clustering task and allocating unknown points into previously determined clusters.

**Keywords:** spatial interpolation; variogram fitting; clustering; bioinspired algorithms

## 1. Introduction

Kriging is a geostatistical interpolation technique that predicts the value of observations in unknown locations based on previously collected data [1]. The kriging error or interpolation error is minimized by studying and modelling the spatial distribution of points already obtained. This spatial distribution or spatial variation is expressed in the form of an experimental variogram.

The experimental variogram can be considered as a graphical representation of the data distribution and also expresses the data variance with the increment of the sampling distance. The variogram is the basis for the application of the kriging method. Thus, the kriging process is defined into three main steps. First, the experimental variogram is calculated. In the sequel, the theoretical variogram is modeled to represent the experimental variogram. Finally, the value of a new point is predicted using the built theoretical model [2].

Despite advances in geostatistics and kriging areas, the task of modelling the theoretical variogram remains a challenge (step 2), which is mainly responsible for the interpolation accuracy. In order to model the theoretical variogram, it is necessary to estimate its parameters, which can be defined as an optimization problem [3].

In recent years, artificial intelligence techniques have been used to improve the kriging process as shown in [4–9]. However, it is still a challenge to determine which method is better suited for a given database. As stated in [6] and applied in [7], bio-inspired algorithms, like Genetic Algorithms (GA), are suitable to help define the theoretical variogram parameters. Furthermore, these kinds of algorithms do not require a single initial seed value as input, but rather a minimum and maximum interval, different from classical methods such as Gauss–Newton [3] and Levenberg–Marquardt [10].

In an automated setting, these seeds and intervals must be determined based on data; however, this is a hard task since there is no well-defined heuristic to initialize them. In [11], the authors applied a numerical minimization technique and proposed a heuristic to define the seed in the automatic structure of their method. The research presented in [12] applied the Levenberg–Marquardt optimization technique with the least squares cost function. No heuristic was specified, but the authors tested several initial values in the optimization step. The work described in [13] used an improved linear programming method in combination with the weighted polynomial, as well as the inverse of lag distance as the cost function to select the parameters for the theoretical variogram. In this scenario, we propose a heuristic to define the lower and upper bounds of the GA parameters' intervals.

In [14], the authors proposed a new method for kriging using the K-means clustering technique. The proposal consists in creating subsets of the database and predicting each given point using information from the subset of which it belongs. The proposed method presented better results than conventional kriging approaches. However, the authors used the same theoretical variogram model for all subsets, which could be improved, since the characteristics of each subset are different. In [15], each subset has its own theoretical variogram model for interpolation. On the other hand, the researchers used non-spatial data, which do not have spatial coordinates, and did not detail how the parameters of the theoretical variogram model were defined.

Data clustering techniques present some problems in spatial data, such as cluster overlapping. As stated in [16], this behavior of creating spatially scattered clusters (cluster overlapping) is undesirable for many geostatistical applications, since it impacts the properties of spatial dependency over the study domain, and it is important to maintain the original characteristics of the database. Recently, some researchers seek to solve the cluster overlapping problem by developing methods that guarantee spatial contiguity where the groups created are uniform. For instance, in [14], a normalization factor was used to minimize this problem. The method proposed in [17] presented a new hierarchical clustering approach by applying weights to spatial and target variables, and Reference [16] proposed a spectral clustering method to maintain the property of spatial contiguity. For this problem, we proposed a methodology using K-means clustering improved by the K-Nearest Neighbor (KNN) classifier.

Another problem is that, when including coordinate variables in the clustering process, it does not guarantee the stationary hypothesis [16]. Kriging requires that this hypothesis is satisfied [1] that statistical properties such as mean, variance, among others, are all constant over the spatial domain; otherwise, we have the phenomenon called trend. According to [18], the trends removal procedure (also called detrending) ensures the stationary hypothesis on data. We carried out experiments with and without applying the detrending step in order to compare the results.

An important issue that has not been explored in the related works is when a new point in the spatial space needs to be interpolated. Since it must belong to a group, a mechanism to attach this new point to a cluster is needed. We explored this problem by proposing the application of the KNN algorithm to perform this allocation task.

In this context, a new methodology is proposed to improve kriging estimates, using concepts of preprocessing, data clustering, and optimization methods. In our approach, for comparison purposes,

both K-means and Ward-Like Hierarchical [17] clustering procedures were used to separate the data set in groups by similarity. The KNN algorithm performs the allocation task of new data in their respective cluster, and solves some problems occurred in the clustering step of the K-means method, such as cluster overlapping. In order to analyze the impacts on the stationary hypothesis, we applied the Mann–Kendall non-parametric test [19] to evaluate the trend phenomenon. Finally, genetic algorithms were used to model a specific theoretical variogram to each group found in the clustering step, with the definition of the parameters' bounds based on data of each group. Although GA is not considered nowadays as the state-of-the-art meta-heuristics, this technique was chosen in order to serve as a baseline for the proposed model, and also because it was applied in similar kriging methodologies, as can be seen in [5,7,8,20]. For instance, in [20], the authors concluded that particle swarm optimization and genetic algorithms are statistically equivalent in the task of estimating the kriging parameters. The proposed methodology was evaluated using two publicly available databases and the results compared with other optimization techniques presented in the literature.

The remainder of the paper is organized as follows. Section 2 presents the theoretical background. Section 3 describes the proposed methodology steps. Section 4 reports the used databases and the experimental setup. The baseline results are discussed in Section 5. Finally, Section 6 summarizes our conclusions and addresses future works.

## 2. Background

This section presents a brief explanation of the main concepts discussed in this paper.

### 2.1. Kriging

Kriging is an interpolation technique widely used in geostatistics to predict spatial data. This method takes into account the characteristics of regional variables autocorrelation. These variables have some spatial continuity, which allows the data obtained by sampling of specific points to be used to parameterize the prediction of points where the value of the variable is unknown [1].

Let $Z$ be a set of observations of a target variable (response variable) denoted as $\{z(s_1), z(s_2), \ldots, z(s_N)\}$, where $s_i = (x_i, y_i)$ is a point in a two-dimensional geographical space; $x_i$ and $y_i$ are its coordinates (primary locations); and $N$ is the number of observations.

Values of the target variable at some new location $s_0$ can be derived using a spatial prediction model. The standard version of kriging is called ordinary kriging (OK), where the predictions are based on the model:

$$\hat{z}_{OK}(s_0) = \sum_{i=1}^{N} w_i(s_0) \cdot z(s_i) = \lambda_0^T \cdot \mathbf{z}, \tag{1}$$

where $\lambda_0$ is a vector of kriging weights $(w_i)$, and $\mathbf{z}$ is the vector of $N$ observations at primary locations.

Thus, in order to estimate the weights, we calculate the semivariances $\gamma(h)$ based on the differences between the neighboring values:

$$\gamma(h) = \frac{1}{2}E[(z(s_i) - z(s_i + h))^2], \tag{2}$$

where $z(s_i)$ is the observation of the target variable at some point location, and $z(s_i + h)$ is the observation of the neighbor at a distance $s_i + h$.

Suppose that there are $N$ point observations, this yields $N \times (N-1)/2$ pairs for which a semivariance can be calculated. If we plot all semivariances versus their separation distances, a variogram cloud is produced. For an easier visualization of this variogram cloud, the values are commonly averaged for a standard distance called "lag". If we display such averaged data, then we get the standard experimental variogram, which can be seen in Figure 1.

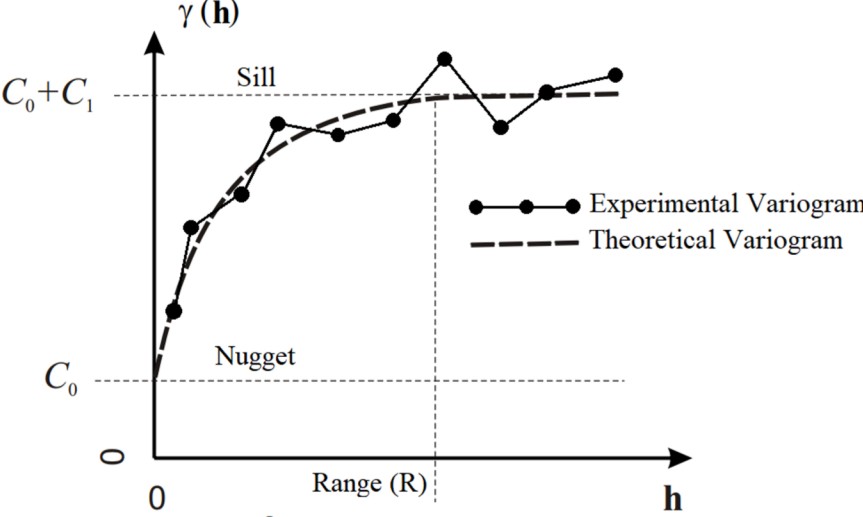

**Figure 1.** Example of a final variogram model.

Once we calculate the experimental variogram, we can fit it using a theoretical model, such as linear, spherical, exponential, Gaussian, among others. The variograms are commonly fitted using a cost function (e.g., weighted least squares [21]). Hence, the main objective is to minimize this cost function. In this work, in order to simplify the experiments, we only use the matern theoretical model, which is is given by

$$\gamma(h) = C_0 + C_1 \left\{ 1 - \frac{1}{2^{v-1}\Gamma(v)} \left( \frac{h}{R} \right)^v K_v \left( \frac{h}{R} \right) \right\}, \tag{3}$$

where $R$ is the practical range and it is equal to the distance for which $\gamma(h) = 0.95(C_0 + C_1)$ [22]. Sample locations separated by distances closer than the range are spatially auto-correlated, otherwise they are not; $C_0$ is the nugget effect, which can be attributed to measurements errors or spatial sources of variation at distances smaller than the sampling interval; $C_0 + C_1$ is the sill, which is the value that the model attains at the range $R$; $v$ is a smoothness parameter called kappa; $K_v$ is a modified Bessel function [23]; and $\Gamma(v)$ is a factorial function of complex numbers.

Once we have estimated the theoretical model, we can use it to derive semivariances at all locations and solve the kriging weights. The ordinary kriging (OK) weights are solved multiplying the covariances:

$$\lambda_0 = \mathbf{C}^{-1} \cdot \mathbf{c_0}; \quad C(|h| = 0) = C_0 + C_1, \tag{4}$$

where $\mathbf{C}$ is the covariance matrix derived for $N \times N$ observations and $\mathbf{c_0}$ is the vector of covariances at a new location. Note that the $\mathbf{C}$ is in fact a $(N + 1) \times (N + 1)$ matrix if it is used to derive kriging weights, since one extra row and column are used to ensure that the sum of weights is equal to one:

$$\begin{bmatrix} C_{(s_1,s_1)} & \cdots & C_{(s_1,s_N)} & 1 \\ \vdots & \ddots & \vdots & \vdots \\ C_{(s_N,s_1)} & \cdots & C_{(s_N,s_N)} & 1 \\ 1 & \cdots & 1 & 0 \end{bmatrix}^{-1} \cdot \begin{bmatrix} C_{(s_0,s_1)} \\ \vdots \\ C_{(s_0,s_N)} \\ 1 \end{bmatrix} = \begin{bmatrix} w_1(s_0) \\ \vdots \\ w_N(s_0) \\ \varphi \end{bmatrix}, \tag{5}$$

where $\varphi$ is the *Lagrange multiplier*. After calculating the weights, the prediction is then given by Equation (1).

When the experimental variogram is distinct for two or more directions, we have an anisotropic phenomenon [1], as can be seen in the ellipse drawn in Figure 2. The ellipse represents the area from which data would be considered in kriging process. The anisotropy is calculated considering a certain

angle from 0 to 180 degrees, which represents the azimuthal direction clockwise on the major direction, and a factor given by

$$AnisotropyFactor = \frac{a_2}{a_1},$$ (6)

where $a_1$ and $a_2$ are the biggest and smallest radius of the ellipse, respectively. This factor varies between 0 and 1, with 1 being an isotropic model. Therefore, in case of anisotropy, five parameters are used to estimate the theoretical variogram model: nugget, sill, range, angle, and the anisotropy factor.

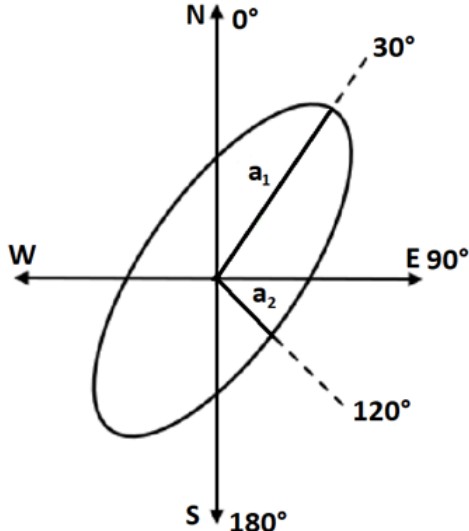

**Figure 2.** Example of anisotropy.

*2.2. Stationary Hypothesis*

Given a set of $N$ values $z(s_i)$, the hypothesis will be intrinsic (or stationary) if it follows two conditions. The first condition requires that the expected value $E\{z(s_i)\}$ exists and does not depend on the position $s_i$. This is mathematically written as

$$E\{z(s_i)\} = m.$$ (7)

Regarding the second condition, the variance of the increment $[z(s_i) - z(s_i + h)]$ is finite and does not depend on the position $s_i$. This can be written as

$$VAR[z(s_i) - z(s_i + h)] = E[z(s_i) - z(s_i + h)]^2.$$ (8)

Trends are patterns or noise on data, usually expressed as an upwards or downwards shift over time or space. Note that both conditions are not met with the presence of trend. This is because the mean value $m$ would depend on position $s_i$ and the quantity $VAR[z(s_i) - z(s_i + h)]$ could be infinite, and it would also depend on the position in the field [18].

The task of removing trends (or detrending) can be done by fitting a trend surface to the values by, for example, the least squares method, and then subtracting the value of the trend surface function from the original values, becoming a new residual variable [18].

Detrending the variables is a requirement for non-stationary values to be under the stationary hypothesis (for stationary geostatistical techniques). After detrending, the analysis follows its normal way by calculating the experimental variogram, fitting a model, and applying the kriging estimates. At the end, it is possible to add back the values of the trend surface to obtain the original values.

### 2.3. KNN

K-Nearest Neighbor (KNN) [24] is a supervised machine learning algorithm used for both classification and regression problems. It is a non-parametric approach that uses training data directly for classification. More specifically, the KNN algorithm classifies a new point based on the training set points that are close to it.

Given a training data set $\{(\mathbf{s}_1, q_1), \ldots, (\mathbf{s}_N, q_N)\}$, with $N$ points, each point $(\mathbf{s}, q)$ consists of a vector $\mathbf{s} \in \mathbb{R}^L$ and a label $q \in \{1, \ldots, Q\}$. Let $\mathbf{s}_0 = (p_1, \ldots, p_L)$ be a new point not yet classified (i.e., without label). In order to classify this new point, the KNN algorithm calculates the distance between $\mathbf{s}_0$ and the other points in the training set using a measure of similarity. The $K$ nearest points (i.e., with smaller distances) in relation to $\mathbf{s}_0$ are then stored. In the sequel, it is verified which is the most frequent label among the $K$ neighbors, and this elected label $q$ is associated with the new point.

A well-known measure of similarity and also used in this work is the Euclidean Distance, which is defined by

$$d(\mathbf{s}, \hat{\mathbf{s}}) = \sqrt{\sum_{i=1}^{L}(p_i - \hat{p}_i)^2},$$
(9)

where $p_i$ and $\hat{p}_i$ are elements of vectors $\mathbf{s}$ and $\hat{\mathbf{s}}$, respectively.

### 2.4. K-Means

K-means [24] is one of the simplest unsupervised learning algorithms that solves the clustering problem. This clustering algorithm partitions the database into $U$ clusters, where the value of $U$ is provided by the user.

The K-means algorithm starts by initializing a set of $U$ centroids, one for each cluster. A widely-used initialization method is the random selection among the points in the database. Each point is then associated with the nearest centroid based on a measure of similarity (e.g., Euclidean Distance) in order to build the groups. After that, the centroids are recalculated. The "new" centroid is the average of the "old" cluster points. This process is repeated until the centroids are no longer modified.

### 2.5. Genetic Algorithms

Genetic algorithms (GA) [25] are a group of optimization methods inspired by natural biological evolution. A GA is composed by a population of individuals that represent its search space. In other words, an individual is a possible solution to the problem, and it is coded as a finite length vector in terms of an alphabet.

This algorithm begins with a population of randomly generated individuals. Then, each individual is evaluated by the fitness function. This function verifies how well the individual solved the problem in analysis. After that, the selection operator chooses the individuals who will pass to the next generation. Finally, the crossover and mutation operators are applied. The algorithm runs until a stopping criterion is fulfilled, such as the number of generations.

### 2.6. Evaluation Metric

The fitness function used in the GA algorithm was obtained by applying the kriging process at each data point (leave-one-out cross-validation) of the training data (90% of the database). Regarding the evaluation metric, the 10-fold cross-validation was applied. For both cases, fitness and evaluation, the interpolation cost function (Equation (10)) was employed [7]. More specifically, the normalized mean squared error (NMSE) index was used as figure of merit and calculated by

$$NMSE_u = \frac{1}{\sigma^2 \cdot n} \sum_{i=1}^{n} [z\hat{O}_K(s_i) - z_{OK}(s_i)]^2,$$
(10)

where $z_{\hat{O}K}(s_i)$ is the predicted value of the target variable obtained by the ordinary kriging method at the hidden point $s_i$; $z_{OK}(s_i)$ is the real value of the target variable at the hidden point $s_i$; $n$ is the total number of points in the cluster $u$; and $\sigma^2$ is the variance of the target variable considering the cluster $u$ data. A lower NMSE indicates a better prediction value. Thus, the NMSE index of the database is given by

$$NMSE = \sum_{u=1}^{U} NMSE_u, \tag{11}$$

where $U$ is the total number of clusters.

It is important to point out that the leave-one-out cross-validation was used only to calculate the GA fitness function. In order to measure the accuracy, we applied the 10-fold cross-validation. Thus, the studied databases were randomly partitioned in 90% for training and 10% for test in each iteration. In all, 10 iterations with different partitions were performed for each number of clusters. The average of these 10 tests was calculated in the end.

## 3. Proposed Methodology

The proposed methodology consists of an automatic process to define the theoretical variogram parameters, aiming to enhance the kriging process as a whole and reduce the dependence that exists today on specialist knowledge. Figure 3 summarizes the proposed methodology, which is divided into four main steps: (i) data preprocessing; (ii) data clustering; (iii) building a model for each subset separately; and (iv) cluster assignment of new data for interpolation. For each number of cluster (1 to 3), this process was repeated 10 times in the 10-fold cross-validation, except for the data preprocessing step. Each iteration was performed using different training and test data sets.

Initially, the data preprocessing step is carried out applying standardization algorithms, treatment of outliers, and data detrending. After that, $U$ data subsets are generated based on the chosen clustering technique. Finally, an optimization algorithm (or a hybrid approach) is used to estimate the best value of the parameters that will define the theoretical variogram model. This parameters set, represented by the vector $\theta^*$, is calculated by the objective function

$$\theta^* = \arg\min_{\theta \in \Theta} M(\theta), \tag{12}$$

where $\theta$ is a parameter set obtained automatically from the experimental variogram data, and $M(\theta)$ represents the result of the cost function for $\theta$. Notice that one model is fitted for each cluster.

Regarding the assignment of new data, a supervised machine learning model (or classifier) is used to choose the best cluster for this unknown point. Hence, the data of the assigned cluster are used to interpolate the new point.

It is important to observe that any algorithm of the proposed methodology can be modified according to the problem to be treated. In other words, the main contribution of this work is to provide a baseline to be followed, bearing in mind that different preprocessing, clustering, classification, and optimization techniques can be applied. Other relevant contributions are the normalization of the cluster groups via KNN, a heuristic to define the GA parameters bounds and a method to classify new points to a cluster.

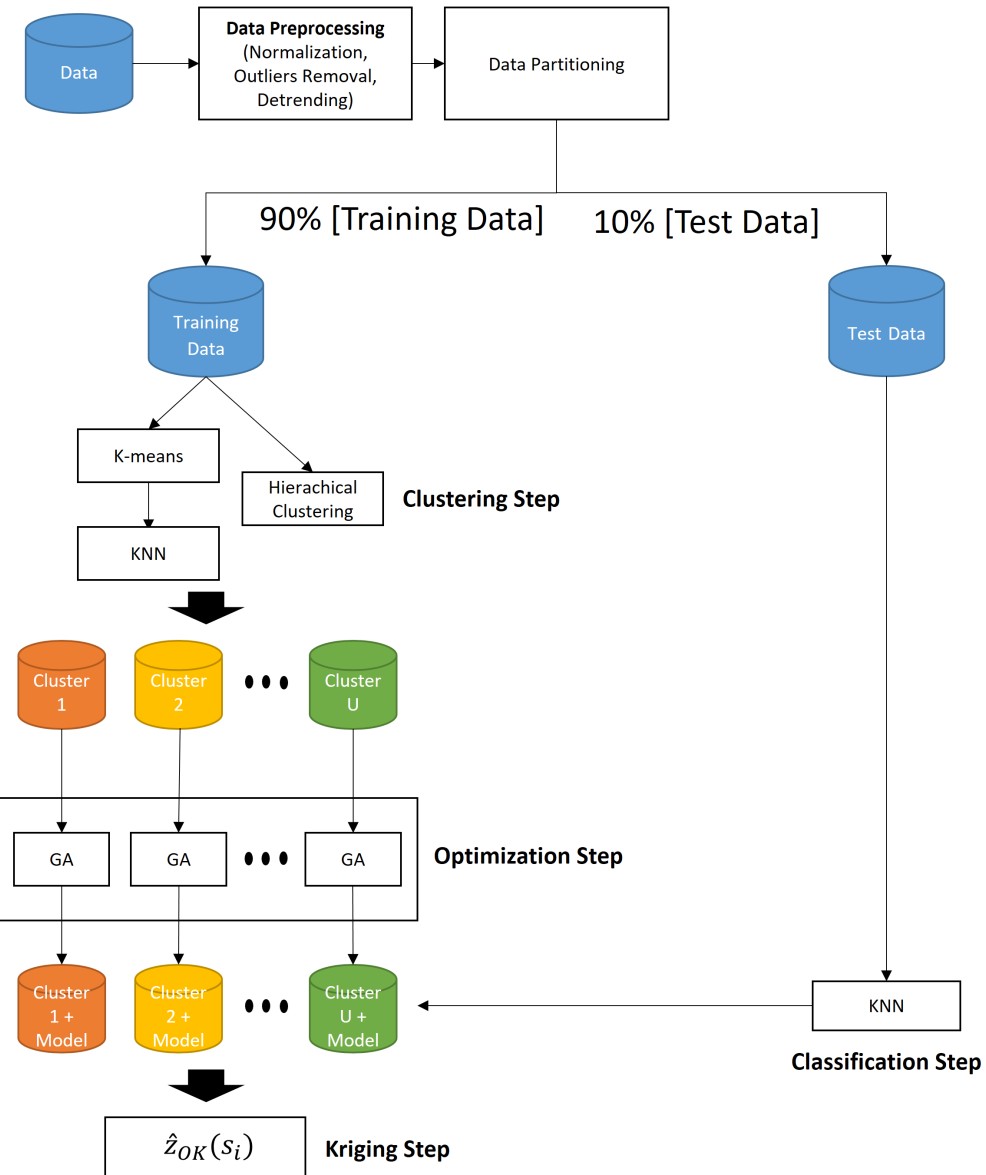

**Figure 3.** The general scheme of data preprocessing, data clustering, and model fitting stages for each cluster separately.

### 3.1. Data Preprocessing

Since outliers can have a big (and negative) impact on the clustering and kriging process [26], in some cases, we need to apply a mechanism to treat these elements [27]. Thus, first, the vectors of primary locations **x** and **y**, as well as the set of observations of the target variable **z**, were normalized between 0 and 1. This procedure is important to ensure that every variable has the same weight in the clustering process and to avoid cluster overlapping. In the sequel, we used the *z*-score test [28] with 99% of confidence to remove the outliers. Finally, we applied a detrending process to ensure the stationary hypothesis on the original data. In our experiments, a second order polynomial function was used to fit the trend surface [18]. It is important to mention that, depending on the nature of the problem, keeping the outliers in data can be a valuable source of information.

## 3.2. Data Clustering

The training data were split into $U$ clusters using the K-means algorithm with the spatial information ($x$ and $y$ coordinates) and the target variable. As shown in [14], the clustering process often results into overlapped clusters. Thus, in order to minimize the overlapping, all data were previously normalized (between 0 and 1), and the KNN algorithm was applied to enhance the data grouping by allocating the current point based on the $k$ neighbors.

The results in the experiments section were obtained using the three nearest neighbors (3-NN) configuration. The number of neighbors were tested from 1 to 5, and the 3-NN setting was responsible for 70.83% of the best results. For example, the black circle highlighted in Figure 4a demonstrates data overlapping, which was reduced with the application of the proposed methodology, as can be seen in Figure 4b.

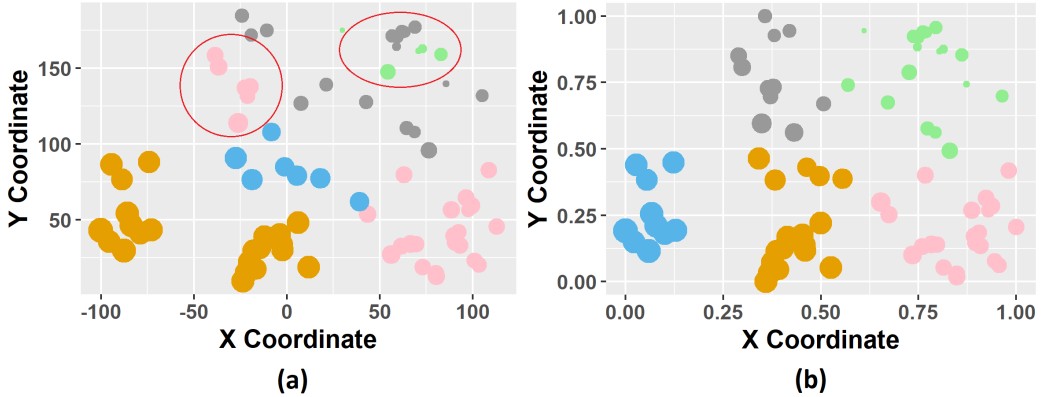

**Figure 4.** Application example of the KNN algorithm in order to minimize the data overlap. (**a**) Before Normalization + KNN; (**b**) After Normalization + KNN.

An implementation of the Ward-Like Hierarchical Clustering algorithm [17], called ClustGeo, was used in the clustering step as an alternative to the K-means + KNN solution. In other words, the idea is to have ClustGeo as a baseline. The ClustGeo algorithm applies "weights" to the spatial coordinates and to the target variable, which we call the alpha parameter. This parameter must be manually tested to find the best spatially contiguity groups. This parameter was set to 0.4 after some initial tests from 0.1 to 0.9.

## 3.3. Optimization

Genetic algorithms were used to optimize (or estimate) the best value of the parameters that will define the theoretical variogram model. Therefore, the GA chromosome was structured with the following parameters: sill, range, kappa, angle, and factor. Kappa is a parameter of the matern variogram model which better suits the GA optimization method, since it can mimic other existing models, like exponential and Gaussian, based on Kappa value.

The chromosome's scheme used in the GA configuration is illustrated in Figure 5. The green blocks represent the active parameters used in the optimization process, and the red ones illustrate the parameters that were not optimized (i.e., their values were previously set). The nugget effect and the number of lags parameters were set to 0 and 10, respectively. Furthermore, GA implementation of [29] was used, which applies tournament selection [25], Laplace crossover [30], and power mutation [31].

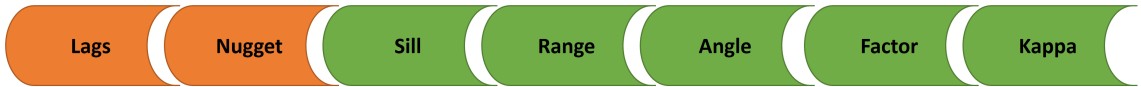

**Figure 5.** The chromosome's scheme used in the genetic algorithm configuration.

*3.4. Classification*

The KNN algorithm was applied to classify the test data points (or new points) into one of the previously built clusters based only on the spatial coordinates. Then, the kriging process was carried out and the error was calculated.

## 4. Experiments

This section presents and discusses the baseline results. The experiments evaluated the proposed methodology effectiveness against other approaches. The adopted implementation of the Ward-Like Hierarchical Clustering algorithm was the R package called "ClustGeo" [17]. All the tests were executed on a computer with an Intel Core i5-4570 3.20 GHz processor (Belém, Pará. Brazil) and 8 GB of RAM. Scripts used in this research can be found in [32], including source codes and database files.

*4.1. Databases*

In summary, two publicly available databases were used to evaluate the proposed methodology, namely: Meuse [33] and Wolfcamp [34]. The number of samples and the target variable of each database are shown in Table 1. The databases were chosen based on their analogous behaviors regarding the trend levels.

**Table 1.** Information about the databases.

| Database | Instances | Target Variable |
|----------|-----------|-----------------|
| Meuse | 155 | Zinc |
| Wolfcamp | 85 | Piezometric Level |

*4.2. Experimental Setup*

The algorithms applied to optimize the parameters of the theoretical variogram are described in Table 2. The main goals are to use GA as a baseline for the proposed model and compare its performance with deterministic methods in order to study the most appropriate approach to the kriging problem. Note that both third-party and proposed algorithms were designed with different cost functions, as well as being executed with and without the anisotropic parameters for comparison purposes.

**Table 2.** List of the optimization algorithms used in the experiments.

| Algorithm | Cost Func. | Anisotropy | Abbreviation |
|-----------|-----------|------------|--------------|
| Proposed GA | Interpolation [7] | Yes | GA |
| Gauss-Newton | Iterative Least Squares | Yes | GN-ILS1 [3] |
| L-Marquadt [10] | Weighted Least Squares | No | LM-WLS [35] |
| Gauss-Newton | Iterative Least Squares | No | GN-ILS2 [3] |

One of the cost functions that evaluates the quality of the selected parameters to be optimized was the weighted least squares (WLS) [10,21], which, according to [7], presents the best results in the adjustment of the theoretical model compared to other cost functions, such as the ordinary least squares and the generalized least squares. More recently, in [3], the iterative least squares (ILS) was

proposed as an enhancement of the WLS cost function. Finally, the proposed GA used the cost function based on the interpolation error (see Equation (10)), as was also employed in [7].

GN-ILS1, GN-ILS2, and LM-WLS are well-known deterministic algorithms that require an initial seed to start the optimization process due to their design. The values presented in Table 3 were then defined according to [3] and [11], where $\sigma^2$ is the variance of the target variable and $d$ is the longest distance between two points considering each of the data clusters separately. The initial value of Kappa was set to 0.5.

**Table 3.** Initial seed for GN-ILS1, GN-ILS2 and LM-WLS algorithms.

| Algorithm | Sill | Range | Angle | Factor | Kappa |
|---|---|---|---|---|---|
| GN-ILS1 | $\sigma^2$ | $d/2$ | 0°, 45°, 90°, and 135° | 1 | 0.5 |
| GN-ILS2 | $\sigma^2$ | $d/2$ | - | - | 0.5 |
| LM-WLS | $\sigma^2$ | $d/2$ | - | - | 0.5 |

Regarding the proposed GA, brute force pre-tests were conducted in order to set a suitable number of generations and reduce the computational cost, in other words, no tuners or heuristics, such as [36–38], were applied to this matter. Hence, the number of generations was set to 20, with 50 individuals in each population, and the crossover and mutation rates were set to 0.9 and 0.1, respectively.

In addition, an interval of possible values must be defined for each variable to be optimized. For this, a heuristic was created to automatically set the lower and upper bounds in order to cover all possibilities, as described in Table 4.

**Table 4.** Heuristic applied to the proposed GA.

| Algorithm | Bound | Sill | Range | Angle | Factor | Kappa |
|---|---|---|---|---|---|---|
| GA | Lower | 0 | 0 | 0 | 0 | 0 |
|  | Upper | $\sigma^2 \cdot 5$ | $d$ | 180° | 1 | 1 |

Regarding the sill parameter, several upper bound values were tested, and the results are shown in Figure 6. The NMSE index average after five runs was evaluated for all databases using GA without clustering. All the configurations achieved equivalent results, and, for the next experiments, the sill upper bound value was set to $\sigma^2 \cdot 5$ given that a certain stability was found after it.

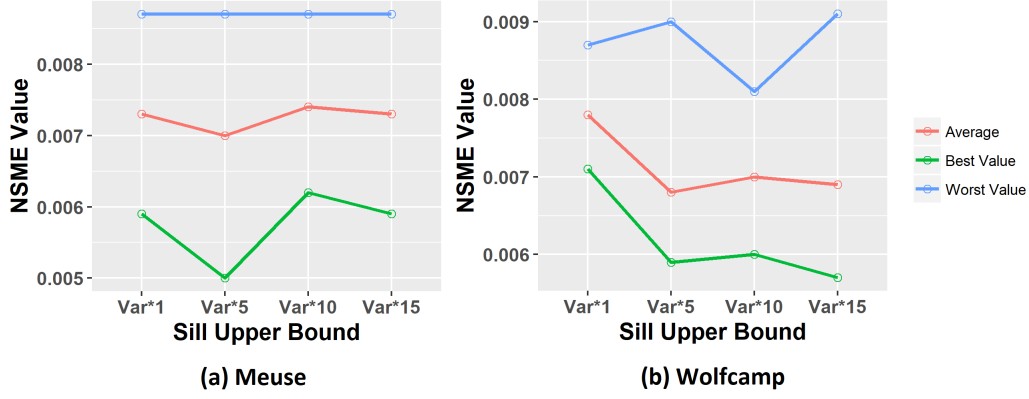

**Figure 6.** Tests with different upper bounds values for the sill parameter in the GA optimization. The *y*-axis represents normalized mean square error (NMSE), and the *x*-axis represents four upper bound values: the target variable variance multiplied by 1, 5, 10, and 15.

## 5. Discussion

The removal of trends is one of the tasks performed in the data preprocessing block. Thus, this first discussion investigates the impact of the detrending process on the stationary hypothesis. The Meuse and Wolfcamp database variograms, before and after the detrending process, are shown on Figure 7. It is possible to see that the detrended data present a more stable behavior than the original ones, especially on the Wolfcamp database, where the variance was exponentially increasing.

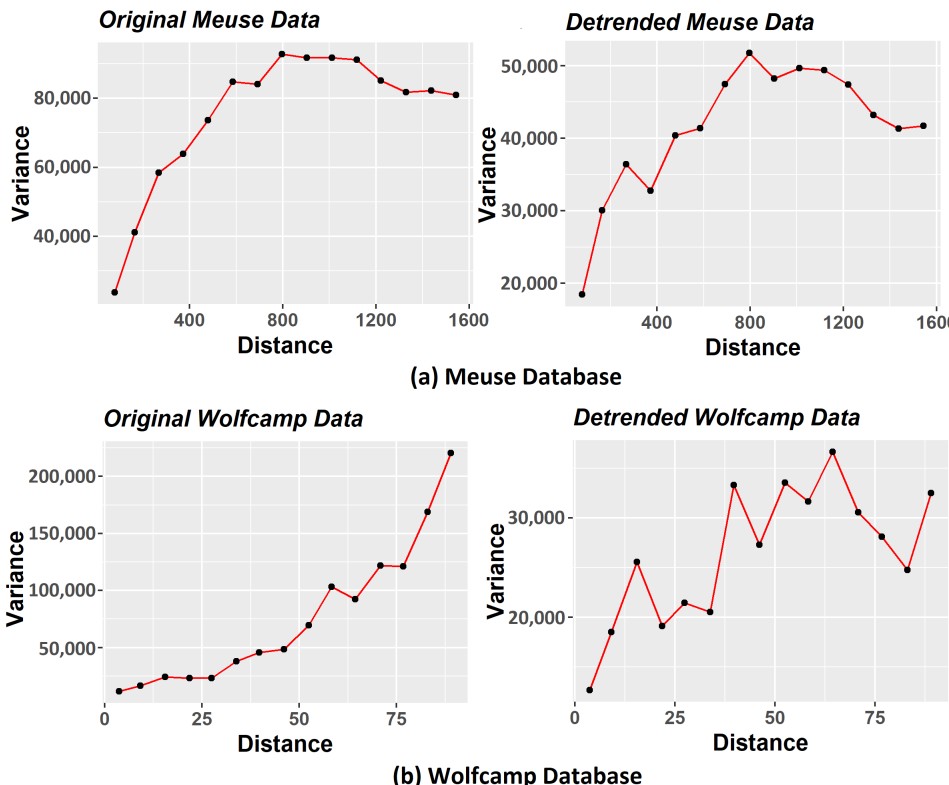

**Figure 7.** Experimental variograms of the Meuse and Wolfcamp databases before and after the detrending process.

The occurrence of trends is more visible in Figures 8 and 9. The inclined line on $x$ and $y$ coordinates on both figures indicates that the mean varies with the distance, also known as trends. The strength of the trend is higher in the Wolfcamp database. After the detrend process, a straight line can be observed, indicating constant mean and stationary data.

Before initiating the clustering step, the proposed hybrid method (K-means + KNN) and the ClustGeo algorithm can maintain the spatial contiguity by adjusting some parameters. However, they do not guarantee the stationary hypothesis of the data clusters created afterward. The results about the occurrence of trends on the groups created by the clustering step are shown in Table 5. A total of 10 executions were performed for each configuration, and the Mann–Kendall test was applied to identify the presence of trends on each cluster. The percentage in Table 5 indicates the number of groups with trends between 10 executions/tests. For both ClustGeo and K-Means+KNN clustering algorithms, the detrending step was beneficial, reducing the occurrence of trends. However, it is important to mention that some clusters, even with the detrending step, still presented the phenomenon, for example, averaging 50% on the Meuse database with 3-cluster configuration (i.e., 15 of the 30 created clusters).

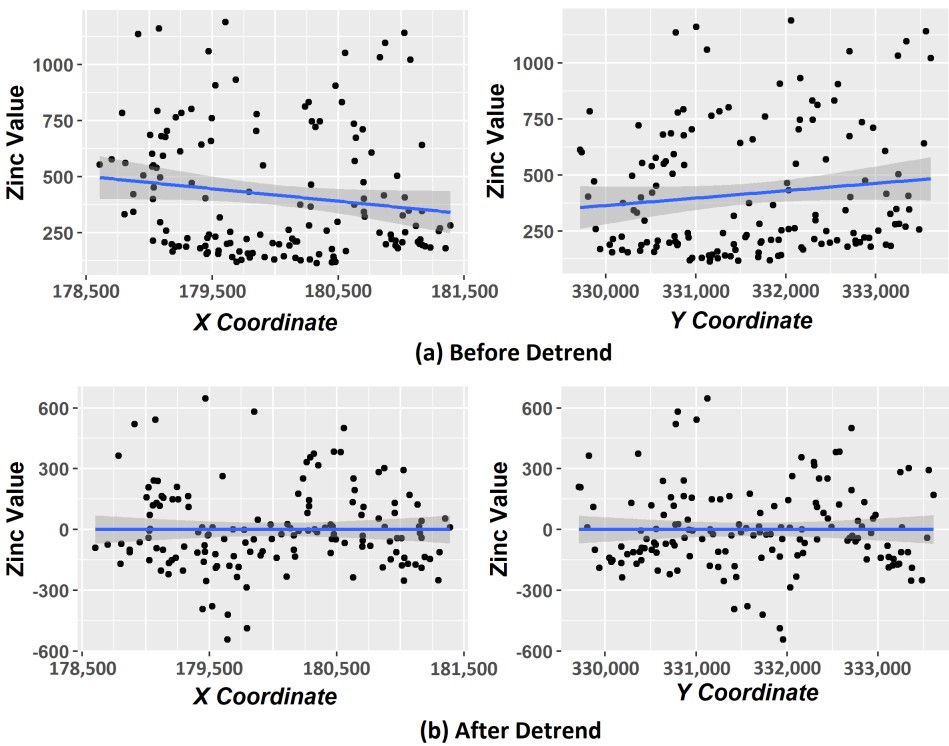

**Figure 8.** Meuse database. Zinc values on *x* and *y* coordinates. (**a**) before the detrending process and (**b**) after the detrending process.

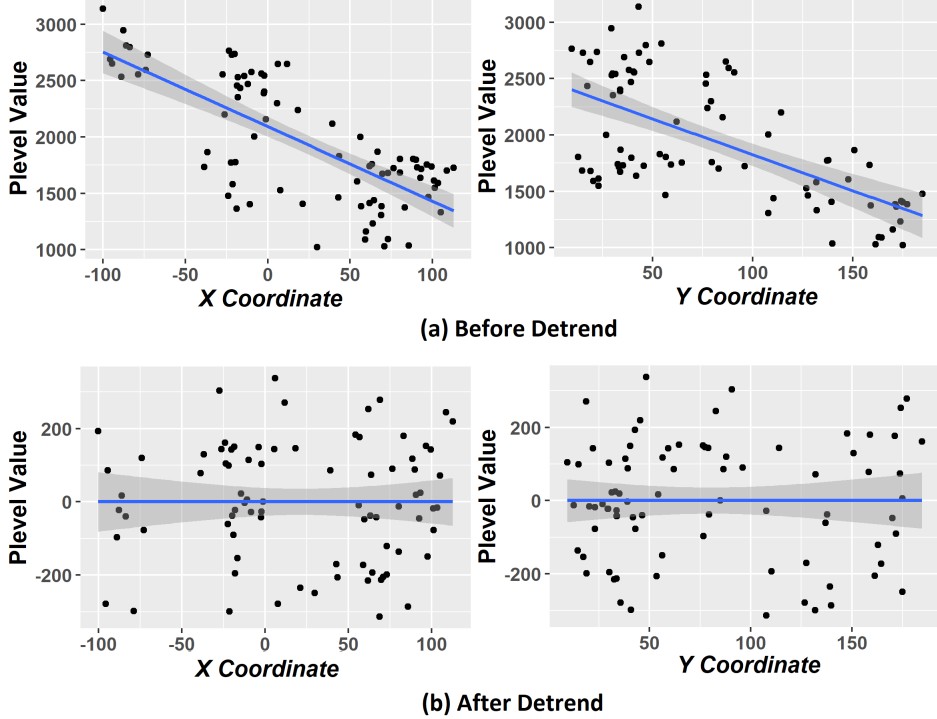

**Figure 9.** Wolfcamp database. Piezometric level values on *x* and *y* coordinates. (**a**) before the detrending process and (**b**) after the detrending process.

**Table 5.** Percentage of occurrence of trends on the created clusters based on the Mann–Kendall index, considering the 1- to 3-cluster configurations.

| | **ClustGeo** | | | |
|---|---|---|---|---|
| | **Clusters** | **1** | **2** | **3** |
| No Detrend | Meuse | 30% | 85% | 70% |
| | Wolfcamp | 100% | 60% | 84% |
| With Detrend | Meuse | 0% | 50% | 47% |
| | Wolfcamp | 0% | 0% | 27% |
| | **K-Means + KNN** | | | |
| | **Clusters** | **1** | **2** | **3** |
| No Detrend | Meuse | 30% | 65% | 44% |
| | Wolfcamp | 100% | 75% | 100% |
| With Detrend | Meuse | 0% | 25% | 50% |
| | Wolfcamp | 0% | 10% | 33% |

After analyzing the influence of trends, we will discuss the main results obtained using the proposed methodology. For this, the treemap information visualization technique was used given the large number of variables employed. Treemap was designed for human visualization of complex tree structures. This technique partitions horizontally and vertically the available space on a non-occlusive squares hierarchy according to the number of tree branches. The higher values (biggest squares) are located on the upper left corner, and lower values (smallest squares) are located on the lower right corner of the structure. Each hierarchy level contains information about one variable, and the principal visual data representations are squares size, color, and label [39,40].

Figure 10 shows the treemap with the results of the experiments performed for the Meuse and Wolfcamp databases. Each square of the treemap represents the average of 10 executions, and its size reflects the NMSE index. The top four results from both databases were highlighted with a red border in the lower right corner of the treemap. It is possible to infer that the best results were reached when the detrending and clustering steps were performed together. In addition, the proposed GA appeared among the best performances. Regarding the clustering techniques, the performance of the proposed hybrid method was compatible with a well-known clustering algorithm, which demonstrates the viability of the solution presented in this work.

One-way ANOVA with repeated measures and paired *t*-Test statistical tests [28] were performed in order to investigate which approaches are statistically significantly better than others with a 95% confidence interval. It is well known that these parametric tests require a data set modeled by a normal distribution. Then, the Shapiro–Wilk test was used for this purpose [28]. Regarding the one-way ANOVA with repeated measures test, if the analysis of variance is significant based on the Greenhouse–Geisser correction [41], it is necessary to use a post hoc test in order to identify which samples are different. Then, the Bonferroni correction [42] was applied for this matter. The statistical analysis results are presented and discussed below. The normality test showed that all variables follow a normal distribution.

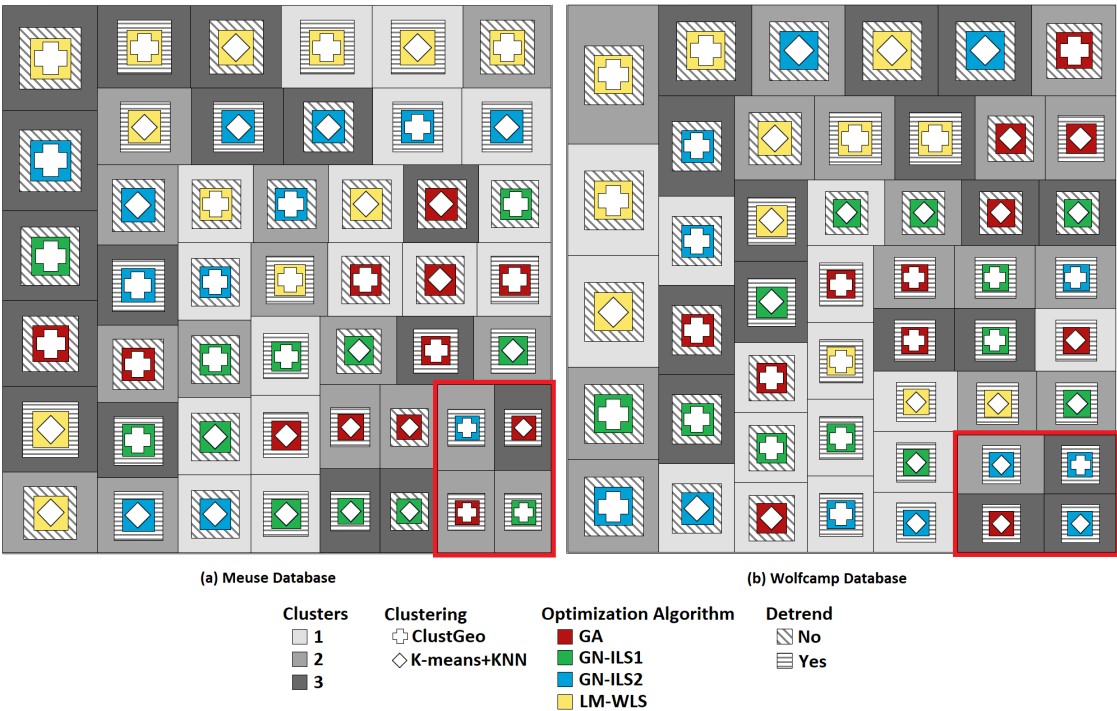

**Figure 10.** Treemap representation of the NMSE index in the various settings tested throughout the experiments using the Meuse and Wolfcamp databases.

## 5.1. Detrending Process

The paired *t*-test was applied with the null hypothesis that the means obtained by the process with and without detrending are equal. The results obtained show that the means differ; in other words, the detrending step has a good effect on the process (i.e., it reduces the NMSE index), considering both databases: $t(479) = 2.658; p = 0.008 < 0.05$ for Meuse and $t(479) = 1.969; p = 0.049 < 0.05$ for Wolfcamp. The descriptive statistics are shown in Table 6.

**Table 6.** Descriptive statistics with and without the detrending step.

|  | **Meuse** | | |
| --- | --- | --- | --- |
|  | **N** | **Mean** | **Std. Deviation** |
| **Without Detrend** | 240 | 0.055 | 0.074 |
| **With Detrend** | 240 | 0.050 | 0.062 |
|  | **Wolfcamp** | | |
| **Algorithm** | **N** | **Mean** | **Std. Deviation** |
| **Without Detrend** | 240 | 0.011 | 0.013 |
| **With Detrend** | 240 | 0.009 | 0.018 |

## 5.2. Data Clustering

Regarding the two evaluated clustering algorithms, Kmeans+KNN and ClustGeo, the paired *t*-test indicated that the means do not differ significantly, considering both databases: $t(239) = 0.607; p = 0.544 > 0.05$ for Meuse and $t(239) = 1.241; p = 0.216 > 0.05$ for Wolfcamp. We can concluded that the proposed strategy was equivalent when compared to another strategy well-known in the literature. The descriptive statistics are shown in Table 7.

**Table 7.** Descriptive statistics for the clustering algorithms.

| Meuse | | | |
|---|---|---|---|
| **Algorithm** | **N** | **Mean** | **Std. Deviation** |
| **ClustGeo** | 240 | 0.054 | 0.066 |
| **K-Means+KNN** | 240 | 0.051 | 0.070 |
| **Wolfcamp** | | | |
| **Algorithm** | **N** | **Mean** | **Std. Deviation** |
| **ClustGeo** | 240 | 0.011 | 0.021 |
| **K-Means+KNN** | 240 | 0.009 | 0.007 |

The one-way ANOVA with repeated measures test reported that the means differ significantly when the number of clusters is increased, considering both databases: $F(1.712, 272.219) = 5.695; p = 0.006 < 0.05$ for Meuse and $F(1.179, 187.442) = 3.822; p = 0.045 < 0.05$ for Wolfcamp. The descriptive statistics and the post hoc test results are described in Tables 8 and 9, respectively. Based on the significance calculated by the Bonferroni test, we can conclude that, on average, the configurations with data clustering were outperformed by the 1-cluster configuration (i.e., without data clustering). This result was not expected. One hypothesis for that is the presence of trends on the groups created by the clustering step, even after the data have been submitted to the detrending process, which does not happen when we use only one cluster, as can be seen in Table 5.

**Table 8.** Descriptive statistics considering the number of clusters.

| Meuse | | | |
|---|---|---|---|
| **#Cluster** | **N** | **Mean** | **Std. Deviation** |
| **1** | 160 | 0.040 | 0.034 |
| **2** | 160 | 0.056 | 0.080 |
| **3** | 160 | 0.063 | 0.079 |
| **Wolfcamp** | | | |
| **#Cluster** | **N** | **Mean** | **Std. Deviation** |
| **1** | 160 | 0.007 | 0.004 |
| **2** | 160 | 0.010 | 0.026 |
| **3** | 160 | 0.012 | 0.009 |

**Table 9.** The Bonferroni test results with different number of clusters.

| Meuse | | | |
|---|---|---|---|
| **#Cluster** | **Mean difference** | **Std. error** | **Sig.** |
| **1-2** | 0.016 | 0.006 | 0.026 |
| **1-3** | 0.023 | 0.007 | 0.002 |
| **2-3** | 0.007 | 0.008 | 1.000 |
| **Wolfcamp** | | | |
| **#Cluster** | **Mean difference** | **Std. error** | **Sig.** |
| **1-2** | 0.003 | 0.002 | 0.373 |
| **1-3** | 0.005 | 0.001 | 0.000 |
| **2-3** | 0.002 | 0.002 | 1.000 |

*5.3. Optimization Algorithms*

Statistically, the adopted optimization algorithm affects the NMSE index, as reported by the one-way ANOVA with a repeated measures test, considering both databases:

F(1.041,123.907) = 43.374; $p = 0.000 < 0.05$ for Meuse and $F(1.032, 122.840) = 3.956; p = 0.048 < 0.05$ for Wolfcamp. The descriptive statistics and the post hoc test results are described in Tables 10 and 11, respectively. According to the Bonferroni post hoc test, on average, the algorithms that do not use anisotropy parameters to estimate the theoretical variogram model (GN-ILS2 and LM-WLS) were outperformed by the ones that use it (GA and GN-ILS1), especially when we analyze the Meuse database. This result was expected. The additional parameters included in GA and GN-ILS1 algorithms (factor and angle) are important to capture the zonal anisotropy, which can be intensified in the clustering step (2 or 3 clusters).

**Table 10.** Descriptive statistics considering the optimization algorithms.

| | | Meuse | |
|---|---|---|---|
| **Algorithm** | **N** | **Mean** | **Std. Deviation** |
| **GA** | 120 | 0.034 | 0.024 |
| **GN-ILS1** | 120 | 0.034 | 0.025 |
| **GN-ILS2** | 120 | 0.040 | 0.031 |
| **LM-WLS** | 120 | 0.103 | 0.114 |
| | | **Wolfcamp** | |
| **Algorithm** | **N** | **Mean** | **Std. Deviation** |
| **GA** | 120 | 0.008 | 0.005 |
| **GN-ILS1** | 120 | 0.008 | 0.004 |
| **GN-ILS2** | 120 | 0.009 | 0.005 |
| **LM-WLS** | 120 | 0.014 | 0.030 |

**Table 11.** The Bonferroni test results with different optimization algorithms.

| | Meuse | | |
|---|---|---|---|
| **Algorithm** | **Mean difference** | **Std. error** | **Sig.** |
| **GA - GN-ILS1** | 0.000 | 0.001 | 1.000 |
| **GA - GN-ILS2** | 0.006 | 0.002 | 0.001 |
| **GA - LM-WLS** | 0.070 | 0.010 | 0.000 |
| **GN-ILS1 - GN-ILS2** | 0.006 | 0.001 | 0.000 |
| **GN-ILS1 - LM-WLS** | 0.070 | 0.010 | 0.000 |
| **GN-ILS2 - LM-WLS** | 0.064 | 0.010 | 0.000 |
| | **Wolfcamp** | | |
| **Algorithm** | **Mean difference** | **Std. error** | **Sig.** |
| **GA - GN-ILS1** | 0.001 | 0.000 | 0.007 |
| **GA - GN-ILS2** | 0.000 | 0.000 | 1.000 |
| **GA - LM-WLS** | 0.005 | 0.003 | 0.338 |
| **GN-ILS1 - GN-ILS2** | 0.001 | 0.000 | 0.004 |
| **GN-ILS1 - LM-WLS** | 0.006 | 0.003 | 0.183 |
| **GN-ILS2 - LM-WLS** | 0.005 | 0.003 | 0.435 |

*5.4. Kriging Maps*

The kriging prediction and variance maps were generated with and without the detrending step, for both databases, as can be seen in Figures 11 and 12. The maps were generated with parameters of the best result obtained considering all the executions performed in the tests, which are K-Means + KNN with 3-clusters for the Meuse database and ClustGeo with 2-clusters for the Wolfcamp database. GA was the employed optimization algorithm for both. We can observe that the action of separating the original data in different data sets did not impair the spatial continuity of the maps. It is also important to note that, in the proposed methodology, the anisotropy parameters (angle and factor) are defined for each cluster individually; then, if there is zonal anisotropy, it will be modeled correctly.

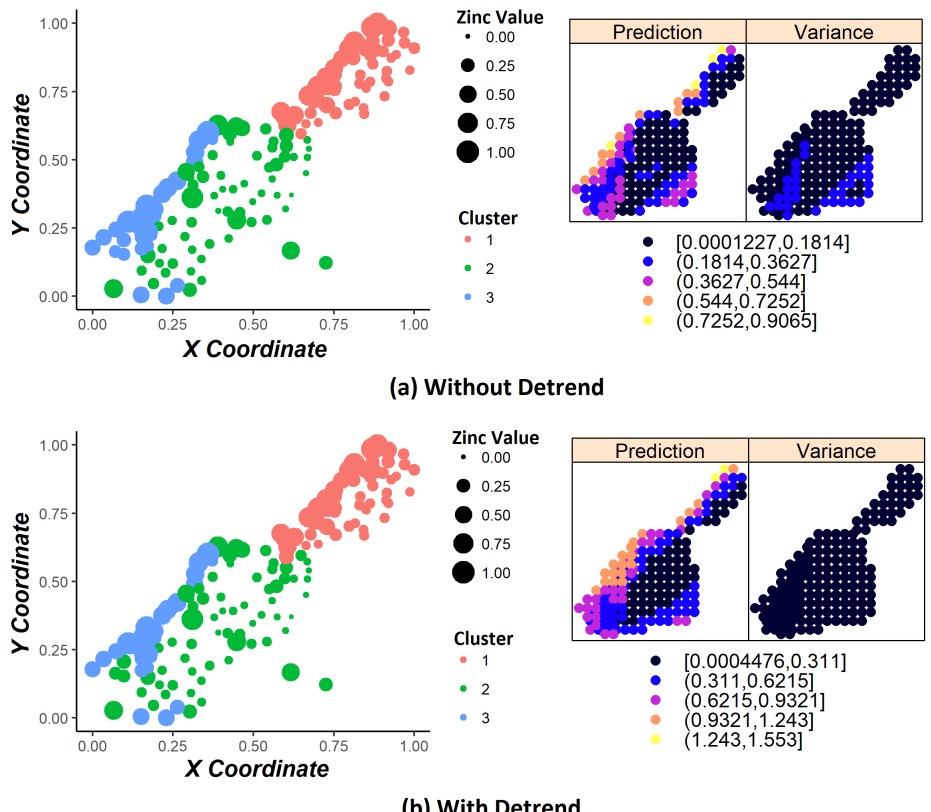

**Figure 11.** Kriging maps (prediction and variance) of the Meuse database.

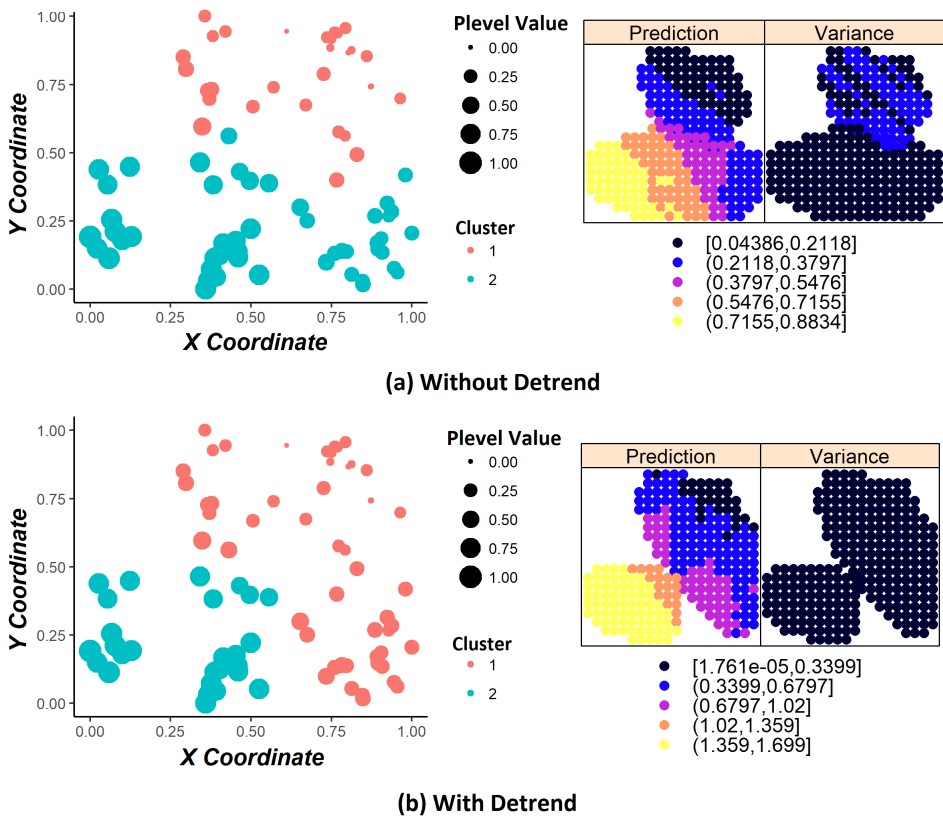

**Figure 12.** Kriging maps (prediction and variance) of the Wolfcamp database.

## 6. Conclusions

Variogram modelling is still a challenge for most geostatistical applications. The results obtained with the proposed methodology using data clustering, evolutionary techniques, and anisotropic parameters achieved lower error rates compared to traditional approaches in the kriging process. In summary, the main contributions of this work are:

- A hybrid configuration (K-means + KNN) was adopted in order to decrease the effect of the cluster overlapping problem.
- One genetic algorithm model was built for each cluster, unlike previous works [14] that provide only one model for all clusters.
- Automatic estimation of the parameters that determine the theoretical variogram according to the database features, which mitigates the need for expert knowledge.
- Clustering the data and estimating specific variogram models to each cluster are procedures that improve the overall accuracy of the kriging process.
- K-Means + KNN and ClustGeo presented distinct performances in different databases. The former algorithm achieved better results on the Meuse database and the latter one on the Wolfcamp database.
- An initial discussion of the impacts of clustering on stationary hypothesis is presented in this research. Removing the trends from the original database is beneficial to the clusters created afterward. This topic needs further research to guarantee stationary hypothesis automatically.

According to our findings, we conclude that the proposed methodology is valid for producing high-quality variogram models. Nevertheless, it is obvious that further research is needed. Future works include evaluating other clustering algorithms where the stationary hypothesis in guaranteed [16]; investigating the impacts of trends on clustered data; adding figures of metric to measure the quality of the formed groups in order to select the number of clusters automatically; improving the classification of new data into clusters using anisotropy information in the KNN algorithm; and using fuzzy techniques to classify the overlapped points.

Another important issue to be addressed is to evaluate other bio-inspired optimization methods, such as differential evolution [43] and artificial bee colony [44], and use meta-heuristics tuners (CRS-Tuning [36], F-Race [37], and others [38]) to set the control parameters of the applied algorithm.

**Author Contributions:** Conceptualization, C.Y., J.P., B.M., N.N. and J.M.; methodology, C.Y., J.P., B.M., N.N. and J.M.; software, C.Y.; validation, C.Y., J.P., B.M., N.N. and J.M.; formal analysis, C.Y., J.P., B.M., N.N. and J.M.; investigation, C.Y., J.P., N.N. and J.M.; data curation, C.Y.; writing–original draft preparation, C.Y.; writing–review and editing, C.Y., J.P., B.M., N.N. and J.M.; visualization, C.Y., B.M., N.N. and J.M.; supervision, N.N. and J.M.

**Funding:** This research received no external funding.

**Conflicts of Interest:** The authors declare no conflict of interest.

## Abbreviations

The following abbreviations are used in this manuscript:

| | |
|---|---|
| GA | Genetic Algorithm |
| KNN | K-Nearest Neighbor |
| WLS | Weighted Least Squares |
| ILS | Iterative Least Squares |
| NMSE | Normalized Mean Squared Error |

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
