# Peer review of "A New Methodology for Automatic Cluster-Based Kriging Using K-Nearest Neighbor and Genetic Algorithms"

_information, doi:10.3390/info10110357_

Round 1

Reviewer 1 Report

The authors proposed a new methodology to help automate the kriging process through the estimation of theoretical variogram parameters. The methodology and the performance measurements using two databases were presented and discussed. The impact of the clustering step and genetic algorithms on the model was investigated as well.

The overall organization of the article needs to be improved. Section 4 on the Materials and Methods should be moved before section 2 and 3 to provide the background of the work. Section 4.7 - 4.11 could be moved into a separate section on the proposed methodology. Section 4.1 could be shortened to leave more space for the figures, which are too small to read the embedded text.

a direct comparison of the error rate (NSME) with/w.o the clustering and GA steps are necessary. Figure 5 doesn't show a quantitative comparison and it might be confusing to some readers. 

Author Response

Dear Reviewer,

We sincerely thank you for all the comments about our manuscript. They surely will improve the work and help our research. In the attached file, we address each comment.

"Please see the attachment".

Reviewer 2 Report

The paper describes a new approach to automate the estimation of theoretical variogram parameters of the kriging process. It consists of preprocessing, data clustering, Genetic Algorithm (GA), and the K-Nearest Neighbor classifier. The problem of correctly setting variogram parameters can be seen as an optimization problem for which metaheuristics can be efficiently applied. The work is interesting, but presentation must be improved.

1) Seems that GA control parameters have been set by trial and error. Have been any tuners used (e.g., CRS-Tuning, F-Race, REVAC)?

2) Figures 1-3, 6-7 are too small and hard to read.

3) Order of sections is odd. I would suggest that sections 2 and 3 (“Experiments” and “Discussion”) are after section 4 (“Materials and Methods”).

4) It is not clear why among different metaheuristics GA was selected. Nowadays, GA is not considered as the state-of-the-art metaheuristics. Have been authors experimented also with Differential Evolution (DE) and Artificially Bee Colony (ABC)?

5) The authors compare GA, GN-ILS1, LM-WLS and GN-ILS2. But it is not clear if comparison is fair. It is hard to compare deterministic and stochastic approaches.

6) Results of Table 5 were not discussed in sufficient details.

7) From Figure 5 it is not clear which approaches are statistically significantly better than others.

8) Some references are incomplete (e.g., [26-27]).

9) Typos:
are an group
->
are a group

References used in this review:

Veček et al. 2016: Parameter tuning with Chess Rating System (CRS-Tuning) for metaheuristic algorithms, Information Sciences 372 (2016) 446–469

Birattari et al. 2002: A racing algorithm for configuring metaheuristics, Genetic Evol. Comput. Conf. 2 (2002) 11–18.

Nannen et al. 2007; Relevance estimation and value calibration of evolutionary algorithm parameters, Int. Joint Conf. Artif. Intell. 7, 975–980.

Trindade and Campelo, 2019: Tuning metaheuristics by sequential optimisation of regression models, Applied Soft Computing, Volume 85, December 2019

Črepinšek et al. 2014: Replication and comparison of computational experiments in applied evolutionary computing: Common pitfalls and guidelines to avoid them. Applied Soft Computing, 19 (2014) 161–170.

Author Response

(The authors gave the same response as above.)

Round 2

Author Response

Dear Reviewer,

Thank you again for all the comments about our manuscript. 

Best regards,

Carlos Yasojima

Reviewer 2 Report

My comments have been addressed. The quality of the paper has been improved by adding statistical tests. I have just minor comments which, can be easily addressed. Namely, the authors wrote:

on page 11, “Regarding the proposed GA, brute force pre-tests were conducted in order to set a suitable number of generations and reduce the computational cost, in other words, no tuners or heuristics were applied on this matter.” But, it would be better to provide some references for tuners and heuristics mentioned. So that a reader has a clear understanding what tuners and heuristics referred to. on page 19, “Another important issue to be addressed is to evaluate other bio-inspired optimization methods, such as differential evolution and artificially bee colony, and use meta-heuristics tuners (CRS-Tuning, F-Race and others) to set the control parameters of the applied algorithm.” Again, it would be good if some references for DE, ABC, CRS-Tuning, F-Race) are provided.

There is no need from my side to review this paper again. The revised version can be checked by the handling editor.

Author Response

Dear Reviewer,

Thank you again for all the comments about our manuscript.

References were added as suggested.

Best regards,

Carlos Yasojima.